# Evaluation of Yeast Derivative Products Developed as an Alternative to Lees: The Effect on the Polysaccharide, Phenolic and Volatile Content, and Colour and Astringency of Red Wines

**DOI:** 10.3390/molecules24081478

**Published:** 2019-04-15

**Authors:** Rubén Del Barrio-Galán, Cristina Úbeda, Mariona Gil, Marcela Medel-Marabolí, Nathalie Sieczkowski, Álvaro Peña-Neira

**Affiliations:** 1Department of Agro-Industry and Enology, Faculty of Agronomical Sciences, University of Chile, P.O. Box 1004, Santa Rosa 11315, La Pintana, Santiago, Chile; marionagilicortiella@gmail.com (M.G.); mmedel@uchile.cl (M.M.-M.); apena@uchile.cl (Á.P.-N.); 2Lallemand Inc. Chile y Compañía Limitada, Rosario Norte 407, piso 6, Las Condes, Santiago, Chile; 3Instituto de Ciencias Biomédicas, Facultad de Ciencias, Universidad Autónoma de Chile, Santiago 7500912, Chile; c_ubeda@us.es; 4Instituto de Ciencias Químicas Aplicadas, Inorganic Chemistry and Molecular Materials Center, Universidad Autónoma de Chile, el Llano Subercaseaux 2801, San Miguel, Santiago, Chile; 5Lallemand SAS, 19 rue des Briquetiers, BP 59, 31 702 Blagnac, France; nsieczkowski@lallemand.com

**Keywords:** yeast derivatives, polysaccharides, red wine colour, phenolic compounds, volatile compounds, astringency

## Abstract

Due to the increase of the use of yeast derivatives (YDs) in winemaking to improve the technological and sensory properties in wines, in this work we evaluated the effect of the post-fermentation application of different yeast derivative products on the physical and chemical properties and astringency of red wines during two consecutive harvests. A commercial and two experimental new yeast derivatives were applied at a medium‒high dosage (30 g/hL). The addition of different yeast derivatives in red wine increased the concentration of different polysaccharide fractions and, therefore, the total polysaccharide content, producing a decrease in the duration of the wine astringency perception over time. The use of yeast derivatives could produce an adsorption/clarification and/or protective effect on the phenolic compounds. However, it did not produce an important modification of the colour parameters. An intensification or a lower decrease of the most volatile compound groups was produced, but it depended on the YDs and yeast strain used in fermentation and post-fermentation processes.

## 1. Introduction

Yeast derivatives have been studied more and more in recent years due to the positive effects they can produce in wines during and after alcoholic fermentation. Their use is indicated during the winemaking process mainly to improve the technological (tartrate and protein stability) and sensorial characteristics and to remove some undesirable wine compounds [1]. These products have been proposed for some years as an alternative to traditional ageing via the lees technique, because they can provide the same benefits while avoiding or reducing some of the disadvantages (the release of polysaccharides during ageing on lees is too slow) [2].

Regarding the sensory quality of the wine, the improvements are mainly attributed to the mannoproteins released by the yeast derivatives during wine ageing [1,3,4,5]. In general, yeast derivatives are classified into five groups depending on the process used in the manufacture, the composition, and the degree of purification: inactivated dry yeasts, yeast autolysates, yeast cell walls, yeast extracts [1], and purified mannoproteins.

Generally, there are few commercial preparations based on yeast derivatives with a high degree of purification, mainly because it is a rather laborious and expensive process. For this reason, most of the commercial preparations available on the market for ooenological use are composed of specific inactivated dry yeast (SIDY), yeast autolysate (YA), or yeast cell walls (CW). SIDYs are products that are inactivated by different methods (enzymatic or thermal) and then subjected to a drying process. YAs are obtained after an incubation period of the yeast biomass, with controlled temperature, to favour the release of their own enzymes from the intracellular content and, subsequently, they are inactivated and dried. Finally, the CWs are obtained from YAs by a centrifugation process [1].

Generally, for ooenological applications, these yeast derivatives are selected for their particular characteristics, such as the high content of polysaccharides, mainly mannoproteins or other yeast compounds such as sterols, amino acids, and peptides. SIDYs are commonly used in winemaking as protectors during the rehydration of active dry yeast [6] and to improve the alcoholic [7] and malolactic fermentation [8]. In addition, several studies have described that the mannoproteins released from yeast derivatives may improve several taste characteristics of red wines such as volume and structure as well as decrease astringency and bitterness [9,10,11], and thus they are considered the polysaccharides from yeasts with the highest ooenological interest. This effect is mainly due to the interaction between the mannoproteins and the phenolic compounds of the wine, limiting the self-aggregation of the tannins [12], resulting in more stable polymeric structures that do not interact with the mouth salivary proteins and, therefore improving the sensory characteristics mentioned above and contributing in a remarkable way to contribute to wine mouthfeel properties [13]. Recent studies have shown the ability of YDs to adsorb or interact with the wine phenolic compounds, modifying their sensory characteristics [14,15]. The interaction between polysaccharides and phenolic compounds can also help stabilize the colour of wines due to the formation of more stable polymer pigments that can prevent or reduce the oxidation of wine, as mentioned in several studies [16,17]. Conversely, other studies conclude that this interaction produces a loss of colour in red wines [4,9,11,18,19,20]. 

YDs can also modify the volatility of wine compounds, because they have the capacity to adsorb some wine volatile compounds [3,21]. On the other hand, the application of these products may induce an enrichment of new volatile compounds in wine, which are formed during their processing and could be released into the wines [22].

Astringency is considered a tactile sensation; some authors point out that this phenomenon could be due to a loss or alteration of oral cavity lubrication [23,24]. According to Bennick (2002) [25], the proteins secreted by the parotid gland seem to have the greatest capacity to bind to phenolic compounds. YDs’ polysaccharides could have a lubricating effect and reduce this tactile sensation. Astringency is one of the most important characteristics that define the quality and persistence of red wine [26,27]. Astringency is a complex sensory characteristic, which depends on time and is related to several sensations that can be perceived simultaneously. The time intensity (TI) method has been used in wine to analyse the bitter and astringent sensations caused by phenols [28], the interaction between astringency and sweetness [29], and astringency characterization in commercial wines [30].

Therefore, the objective of this work was to evaluate the effect of the application of different YDs: one commercial SIDY and two experimental new YDs (YA and CW) on the physical and chemical properties and astringency of red wines previously fermented with two different yeasts, which produce different contents of polysaccharides during the fermentation process. The comparative study of polysaccharide release during the fermentation carried out by these yeasts has already been evaluated in another study carried out by our group [31]. For this reason, it is not the objective of this study to compare the two types of yeast. 

## 2. Results and Discussion

### 2.1. Effect of YDs on the Polysaccharide Content

Figure 1 shows the concentration of total polysaccharides and of the different fractions identified and quantified, according to their molecular weight, after the treatment period (2MT) and three months of bottle storage (3MB) in the 2015 harvest. Three different polysaccharide fractions were identified and quantified, and were classified according to their molecular weight: F1 corresponds with the polysaccharide fraction with a high molecular weight (1200‒110 kDa); F2 corresponds with the polysaccharide fraction with a medium molecular weight (110‒20 kDa); and F3 corresponds with the polysaccharide fraction with a low molecular weight (20‒5 kDa).

In general, it was observed that wines treated with the different YDs had a higher concentration of all polysaccharide fractions and total polysaccharides than control wines after the 2MT and 3MB periods for both wines fermented with Lalvin EC1118^®^ and fermented with Uvaferm HPS^®^ (both supplied by Lallemand-Sudamérica (Santiago de Chile, Chile). However, in general, insignificant differences were found between the different YDs studied and only in a few cases the wines treated with YA have a higher content of polysaccharides than the wines treated with SIDY and CW.

The low molecular weight polysaccharide fraction (F3) was the most abundant fraction in all the cases studied, revealing that, in general, all the wines treated with YDs had a higher content than the control wines, and those treated with YA had the highest content.

This effect of the increase of the wine polysaccharides by the addition of YDs, after alcoholic fermentation, has been reported in other studies carried out in red wines of other varieties [2,4,9], but these studies did not report data about the molecular weight of the different polysaccharide fractions. 

In general, it was observed that the content of polysaccharides decreased between the 2MT and 3MB periods, probably due to the filtering process. 

### 2.2. Effect of YDs Application on the Phenolic Content

Figure 2 shows the results obtained in the analysis of total polyphenols, tannins, and anthocyanins, and Table 1 shows the content of the different low molecular weight phenolic groups studied for both types of fermented wines (Lalvin EC1118^®^ and Uvaferm HPS^®^). In general, it was observed that the content of the studied phenolic families decreased or remained stable throughout the study period.

The application effect of the different YDs on the phenolic families depended on the type of YDs and the phenolic group studied. Thus, it was observed that the wines fermented with Lalvin EC1118^®^ and treated with the different YDs presented a similar or higher content of all the phenolic families studied after the 2MT and 3MB periods. In general, similar results were obtained in those wines fermented with Uvaferm HPS. However, it was observed that in some treatments carried out with YDs a decrease in certain phenolic families’ content was also observed but there was no clear trend. 

The trend of the low molecular weight phenolic groups was not clear during the study. Depending on the YDs used, the content was higher or lower than in the control wines. These results may be due to different effects brought out by the YDs: an adsorption effect of phenolic compounds by the polysaccharides released by the YDs, producing a clarification effect; and another protective effect of the phenolic compounds in the wines treated with these products, producing a greater degradation of phenolic compounds in the control wines.

Some studies carried out with different YDs in red wines indicated that their use could modify the phenolic composition, mainly due to the higher release of polysaccharides (mannoproteins) from yeast cell walls. However, the results obtained have been often contradictory, probably due to the different wine composition, as well as the different YDs composition and characteristics used. In some studies it was observed that treatment with YDs decreased the wine’s phenolic content [4,18,31,32]. However, in other studies, the authors observed that the phenolic content of wines treated with YDs was similar to or slightly higher than the control wines [2,33]. In addition, in a study carried out by Watrelot et al. [34] it was observed that the exogenous addition of polysaccharides to Cabernet Sauvignon samples produced an increase in the tannin concentration.

### 2.3. Effect of YDs Application on the Colour of Wines

Table 1 shows the colour intensity (CI) and CIELab parameters of the different wines. The Uvaferm HPS^®^ fermented wines treated with CW and YA presented higher colour intensity (CI) than the control wines and those treated with SIDY. 

This result is well correlated with the different CIELab parameters evaluated, showing that CW- and YA-treated wines presented lower values of L (lightness) and b (blue-yellow chromatic coordinates), and higher values of a (red-green chromatic coordinates) than the control wines. Lower values of b* and higher values of a* indicated higher red-blue tonalities of wines, increasing their CI [35]. For these reasons, the use of these YDs could help to obtain wine with more stable colour pigments. However, these results were not obtained in the same treatments carried out with the Lalvin EC1118^®^ fermented wine.

Results obtained in the last few years in similar studies have shown the great ambiguity of the impact of polysaccharides on colour stability. Some authors reported that wine polysaccharides such as mannoproteins could improve the colour of wines [16,17,33]; others reported that these compounds could lead to a loss of colour [4,9,10,18,20,32]. Even within the same study, where six different YDs were tested, different results were reported: in some cases an increase in CI was observed and in others it was not [2].

### 2.4. Effect of YDs on the Volatile Compounds

A total of 38 volatile compounds were identified in red wines (Table 2 and Table 3). The content of the different chemical groups of volatile compounds was different after the AMLF period as a function of the yeast strain used (but these results were not evaluated statistically).In general, the content of different volatile groups was higher in wines fermented with Uvaferm HPS^®^ than in those fermented with Lalvin EC1118^®^, with the exception of terpenes. With respect to the esters group, the trend between every ester group (ethyl, methyl, acetate, and isoamyl esters) was very similar to the global trend. However, it is important to note that the concentration of isoamyl and acetate esters of wines fermented with Uvaferm HPS^®^ was significantly higher than the wines fermented with Lalvin EC1118^®^ (Figure 3 and Figure 4), mainly in the case of isoamyl and acetate esters (double and triple, respectively; see the Table 2 and Table 3). The trend of the different volatile compound groups studied after 2MT and 3MB periods was not always the same in the two types of wines fermented and, depending on the treatment applied, the content of these compounds increased or decreased (Figure 3 and Figure 4).

In the case of wines fermented with Lalvin EC1118^®^, no significant differences were found in the content of total ethyl esters after the 2MT period, but the wines treated with SIDY and CW had a significantly higher content of acetate esters than the control wines, mainly due to their high content of isoamyl acetate. In addition, the wines treated with CW had a higher content of total fatty acids than the control wines. However, it seems that the period of bottle storage (3MB) produced significant changes, mainly in the wines treated with CW and YA, revealing significantly higher contents of ethyl, methyl and isoamyl esters than in the control wines (see Table 2). Also, treatment with CW increased the concentration of the acetate esters, producing a significant increase of the total esters content of these treated wines at this point of bottle storage. The esters concentration in wines is usually above their perception threshold, which is one of the reasons why they are major contributors to the global aroma of a wine in sensory evaluations [36]. Esters such as ethyl butanoate, hexanoate, and octanoate, which increase after bottle storage with the addition of YDs (see Table 2), have been related to the red-berry aroma in red wine [37]. The same increase observed in esters after the 3MB period was also observed in the total alcohols and acids. Also, the total amount of terpenes increased with all the treatments. 

In the case of wines fermented with Uvaferm HPS^®^, those wines treated with SIDY1 and YA had higher concentrations, after the 2MT period, of the esters from all the chemical groups compared to the control, with the exception of isoamyl esters, where only the wines treated with SIDY1 had higher content than the control wines. These differences were maintained after the 3MB period but only in the case of SIDY1. However, it is important to note that the wines treated with YA also had a higher total isoamyl ester than the controls. Another remarkable fact is the significant increase of fatty acids after 2MT in wines treated with SIDY1 and YA, which had also been observed in the wines fermented with Lalvin EC1118^®^ yeast but only in the wines treated with CW. Some authors have postulated that the higher levels of fatty acids in wines aged with lees are related to the release of these compounds from cell walls during yeast autolysis [38]. Moreover, these results are in accordance with some studies that found a higher content of some fatty acids using ageing treatment with YDs [3,39,40]; for this reason it is possible that the same effect as the ageing on natural lees could have occurred with the use of these products. Other authors have identified fatty acids and their ethyl and methyl esters in several YDs extracts [41], explaining that the great majority seemed to be produced by Maillard reactions, possibly during the thermal processing of these products from yeast sugars and amino acids and/or peptides. In addition, after the 3MB period, the treatment with SIDY1 accounted for the highest concentration of volatile fatty acids. As occurred in the wines fermented with Lalvin EC1118^®^ yeast strain, no significant differences between the control and treated wines were observed in the total alcohols after the 2MT period. However, the treatment with YA produced wines with higher amounts of 3-methyl-1-butanol and octanol than control wines (see the Table 3). Similar results for the total content of fatty acids were found for total alcohols after the 3MB period, showing that the wines treated with SIDY1 had a higher total content than the control and the other treated wines. Conversely to what was observed in the wines fermented with LalvinEC1118^®^, after the 3MB period, the terpenes group did not change considerably.

The adsorption or enhancing effect of the addition of the same YDs on red wines was mentioned by some authors during the ageing and bottle storage periods [3,40]. Rodríguez-Bencomo et al. [40] explained that, in general, the interactions between YDs and some volatile compounds disappeared during bottling, mainly due to the fact that binding between these products and volatile compounds could be reversible and the latter could be released again into the wine. Del Barrio-Galán et al. [3] also observed an interaction effect between the YDs used and volatile compounds of red wines, but an increase of some of these compounds was also found.

The global balance points out that the wines fermented with Lalvin EC1118^®^ presented a higher total amount of volatile compounds during 2MT and after 3MB compared with the control wines. However, when fermented with Uvaferm HPS^®^, after 2MT the treatment with CW decreased the total amount of volatile compounds and after 3MB the wines treated with YA also had decreased amounts.

Therefore, it seems clear that their effect on the volatile compounds depends on the type of product used, because the soluble colloids from yeasts (especially polysaccharides) can affect the perception of aroma substances in opposite ways, either reducing or increasing their volatility, and could be influenced by the multiple factors mentioned above.

### 2.5. Effect of YDs on the Wine Astringency

Figure 5 shows the percentages of dominance over time for the astringency of the different treatments. For the different wines studied, fermented with either Lalvin EC1118^®^ or Uvaferm HPS^®^, it can be seen that the control wines had greater dominance and persistence of astringency at the end of the test with respect to the wines treated with YDs.

Figure 6 shows the duration and the final time of the astringency perception. Statistically significant differences were found for both wines fermented with Lalvin EC1118^®^ and wines fermented with Uvaferm HPS^®^. The control wines obtained a longer duration and greater final time of astringency perception. The time differences in the astringency perception are well correlated with the lower concentrations of polysaccharides present in the control samples, and therefore a greater perception of the astringency over time is also present. No clear difference was observed between the different YDs applied.

Although, in general, the treatments with different YDs did not produce a significant modification of the phenolic compounds content, they had a significant effect on the decrease of the duration of the astringency sensation of the red wines studied. This could be due to the interaction of the polysaccharides released by the YDs and the phenolic compounds of the wine, forming macromolecular structures that remain as stable colloids in wine, as has already been described in some studies [42,43]. According to these studies, polysaccharides and mannoproteins could prevent the tannins’ self-aggregation, forming more stable aggregates that prevent their polymerization and subsequent precipitation. This could contribute to maintaining the lubrication of the oral cavity, decreasing the sensation of dryness and roughness that characterize the astringency attribute.

## 3. Materials and Methods

### 3.1. Winemaking and Experimental Design

The study was carried out on a Carménère wine variety of the 2015 vintage. The wine was made in the Popeta winery, located in the Maipo Valley region of Chile (34° 27′ 3, 35″ (S) and 70° 46′4 2, 17″ (W)). The alcoholic fermentation was carried out in 300-hL stainless steel tanks. The vinification processes were carried out under the routine work conditions established by the winery.

The same volume of must from the same batch of red grapes, located in the same sector of the vineyard, was fermented with two different yeast strains. One batch was inoculated with 20 g/hL of Lalvin EC1118^®^
*Saccharomyces cerevisiae bayanus* and the other batch with 20 g/hL of Uvaferm HPS^®^
*Saccharomyces cerevisiae*. Both yeast strains were supplied by Lallemand-Sudamérica. The alcoholic fermentation was carried out at a controlled temperature between 25 °C and 28 °C. The malolactic fermentation was carried out spontaneously in the winery. The classical ooenological parameters after malolactic fermentation are detailed in Table 4.

The free SO_2_ was adjusted to 30 mg/L after malolactic fermentation was completed and the two different batches of fermented wines were divided into food-grade plastic tanks where different ageing treatments were applied in duplicate (eight tanks × 25 L).

The different treatments were applied after malolactic fermentation (AMLF): control wines (wines without any YDs applied (**C**)); wines treated with a commercial specific inactive dry yeast SIDY (SIDY) named Optilees^®^; wines treated with two new experimental YDs based in a cell-wall fraction (CW) and a yeast autolysate (YA). The treatments were performed in an underground cellar with a constant temperature (15 ± 3 °C).

The YDs were supplied by Lallemand-South America and were applied in a medium‒high dosage (30g/hL) based on the technical recommendations range (20‒40 g/hL). The treatments lasted two months; a weekly “batonnage” was performed during the first month of treatment, and one every two weeks during the second month. After the treatment period the wines were filtered (without clarification treatment) with a cellulose plate filter and bottled, and were stored at a constant temperature (15 ± 3 °C) during three months in bottle.

### 3.2. Reagents and Standards

The standards of gallic, protocatechuic, caffeic, syringic, *p*-coumaric, ferulic, ellagic and caftaric acids, tyrosol, thyptophol, quercetin, myricetin, astilbin, (+)-catechin and (−)-epicatechin, dextrans, and pectins were purchased from Sigma-Aldrich Chemical Co. (St. Louis, MO, USA). Polyethylene membranes of 0.45 μm and 0.22 μm pore size were acquired from EMD Millipore (Billerica, MA, USA). Sodium sulphate (anhydrous), potassium metabisulfite, vanillin (99%), ethyl acetate, diethyl ether, sodium hydroxide, acetic acid, formic acid, sulphuric acid, ethanol, hydrochloric acid and high-performance liquid chromatography (HPLC)-grade acetonitrile, methanol, and ammonium formate were purchased from Merck (Darmstadt, Germany). All the reagents were of analytical grade or higher.

### 3.3. Analytical Methods

The classical ooenological parameters were analysed according to the official methods established by the OIV (2015) [44]. The extraction, determination, and quantification (mg/L) of polysaccharides according to their molecular weight was carried out following the methodology described by Ayestarán et al. [45] using a High-Performance Size Exclusion Chromatography‒Refractive Index Detector (HPSEC-RID).

Total polyphenol index (TPI), total anthocyanins (expressed in mg/L of malvidin-3-glucoside), and total tannins (expressed in g/L of catechin) were analysed according to the methods established by Ribéreau-Gayon et al. [46]. The HPLC-DAD low molecular weight phenolic compounds extraction, determination, and quantification were carried out according to the method proposed by Peña-Neira et al. [47].

Colour intensity and tonality were analysed according to Glories (1984) [48]. CIELab parameters were calculated according to the MSCV^®^ method (Simplified method to determine the colour of wines) developed by the colour group laboratory of the University of La Rioja (Spain) [49].

All the wines were analysed after two months of treatment (2MT) and after three months of bottle storage (3MB). The analysis of volatile compounds was done employing the headspace solid phase micro-extraction method described in Úbeda et al. [50] using 4-methyl-2-pentanol (0.75 mg L^−1^) as internal standard. After that, gas chromatography analysis was carried out using a 7890B Agilent GC system coupled to a quadrupole mass spectrometer Agilent 5977 inert (Agilent Technologies, Palo Alto, CA, USA). The conditions employed were the same as in Ubeda et al. [50].

The sensory analysis was carried out by 14 trained panellists belonging to the Department of Agro-Industry and Enology of the Agronomical Sciences of the Chile University. The evaluation of the different wines was carried out using FIZZ software (Biosystemes, Dijon, France) and according to the conditions established in [9]. The methodologies used to determine the threshold of astringency perception and to train the panellists in the evaluation and characterization of the astringency attribute are described in Medel-Marabolí et al. [51]. This software is a visual tool that allows one to use different sensory analysis methodologies, automation, and data and processing collection [52]. The methodology used was “Time Intensity”, which is a dynamic sensory analysis technique that allows one to observe the progression of a specific sensory attribute over time [53]. In this study, the astringency was evaluated. The intensity of the astringency was evaluated using a 15-cm unstructured scale, where the “0 value” corresponded with the absence of astringency perception and the “15 value” with the maximum intensity of astringency perception. The time evaluation established for each wine was 90 s.

This methodology can be very useful to study the temporal perception of a specific sensory attribute in wine. Generally, according to the literature consulted, the “Time Intensity” methodology is carried out on a small number of attributes or with a limited number of products, where only one attribute is evaluated [53].

### 3.4. Statistical Analysis

The statistical analysis was carried out with the statistical program “InfoSat version 2012p” (FCA-National University of Córdoba, Córdoba, Argentina). Data were analysed using the one-way analysis of variance (ANOVA) followed by the LSD‒Fisher test, which determined statistically significant differences between the means with a level of significance of 95% (*p* < 0.05). The different yeast strains were treated as two independent groups.

## 4. Conclusions

The addition of the YDs produced an increase in the different polysaccharidic fractions and the total polysaccharide content, and it was YA that released the highest amounts after the treatment period, regardless of the fermentation yeast used.

In general, the addition of YDs did not produce a clear trend in the phenolic content of the wines studied, revealing two different effects: adsorption and/or protective of phenolic compounds. In the same way, the results obtained for the colour of wines were quite ambiguous and only in certain cases did we observe an improvement in the red wine’s colour, probably due to the different composition of the YDs, the wine matrix, and the harvest effect.

Two different effects were found in the red wine composition: an intensification of the most volatile compound groups, or a lower decrease of these compounds after the treatment period, mainly after the bottle storage period. However, although the same YDs treatments were used for both types of fermented wines, the results were not always the same. This may be due to the importance of the yeast strain used during the fermentation process, because the initial concentration of the evaluated volatile compounds depends on the fermentation yeast strain.

In general, the addition of different YDs into red wines in the post-fermentation process produced a significant decrease in their astringency perception, mainly due to a lower duration of this attribute in the wines treated with these YDs. This could be correlated with the higher amounts of polysaccharides released by these products, producing wines with a better mouthfeel that are thus more attractive to the wine consumer.

Based on these results, and due to the scarce scientific studies considering the great variety of YDs (commercial and in development), more research should be carried out in this area in order to obtain a better understanding of the action mechanisms of these products and their effect on the sensory profile.

## Figures and Tables

**Figure 1 molecules-24-01478-f001:**
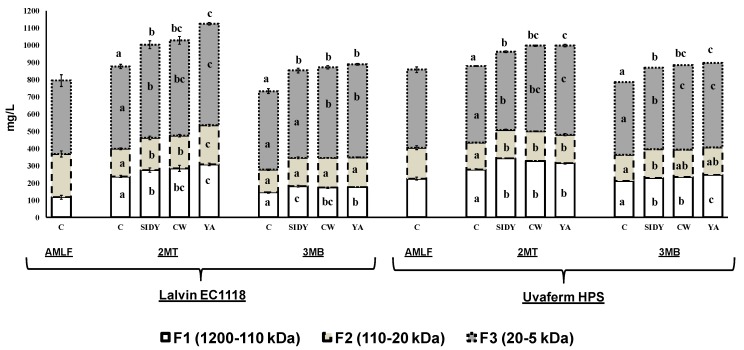
Polysaccharide fractions contents (mg/L ± SD) of the different red wines studied. Columns with different letters indicate statistically significant differences (*p* < 0.05) between the different treatments: AMLF (after malolactic fermentation); 2MT (two months of treatment); 3MB (three months of bottle storage).

**Figure 2 molecules-24-01478-f002:**
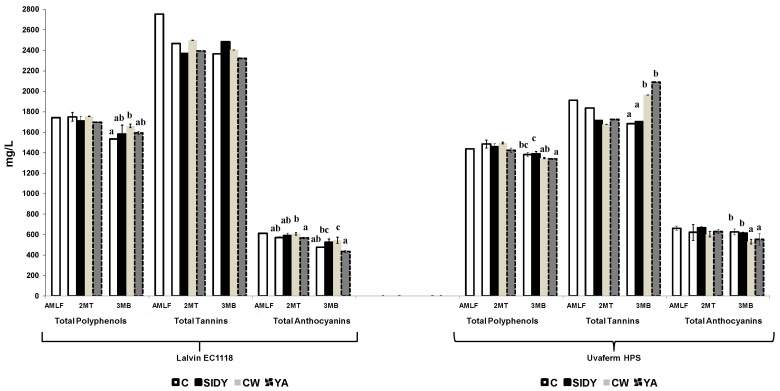
Total polyphenol, tannins, and anthocyanins concentration (mg/L ± SD) of the different wines studied. Columns with different letters indicate statistically significant differences (*p* < 0.05) between the different treatments: AMLF (after malolactic fermentation); 2MT (two months of treatment); 3MB (three months of bottle storage).

**Figure 3 molecules-24-01478-f003:**
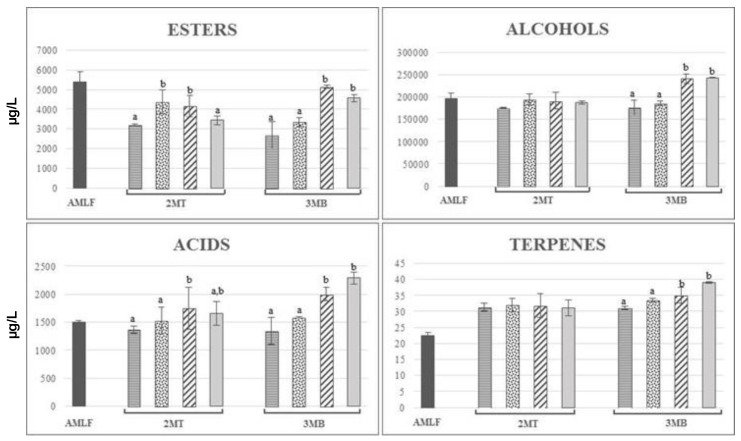
Evolution of the volatile compounds during the treatment and bottle storage in Carménère wines fermented with Lalvin EC1118^®^. Results expressed in µg/L. 
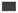
 Control after malolactic fermentation; 
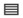
 Control; 
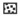
 SIDY; 
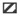
 CW; 
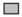
 YA. MT: Months of treatment, MB: Months in bottle. Different letters indicate statistically significant differences (*p* < 0.05) between values.

**Figure 4 molecules-24-01478-f004:**
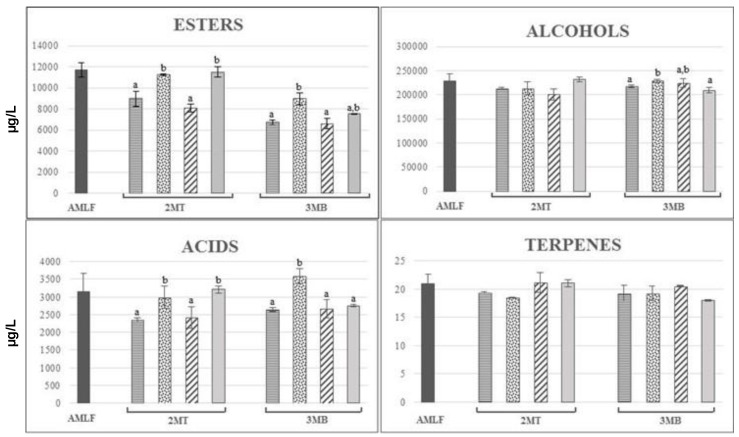
Evolution of the volatile compounds during the treatment and bottle storage in Carménère wines fermented with Uvaferm HPS^®^. Results expressed in µg/L ± SD. 
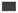
 Control after malolactic fermentation; 
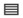
 Control; 
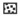
 SIDY, 
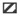
 CW, 
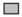
 YA. Columns with different letters indicate statistically significant differences (*p* < 0.05) between the different treatments. 2MT: two months of treatment, 3MB: three months in bottle.

**Figure 5 molecules-24-01478-f005:**
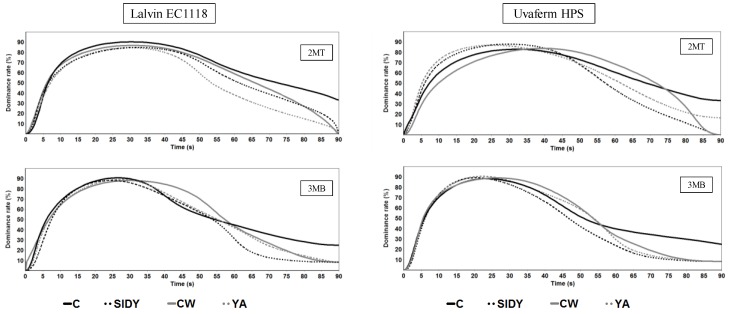
Time intensity profiles of astringency evaluated in the different red wines studied. (s): seconds; (%): percentage of dominance rate.

**Figure 6 molecules-24-01478-f006:**
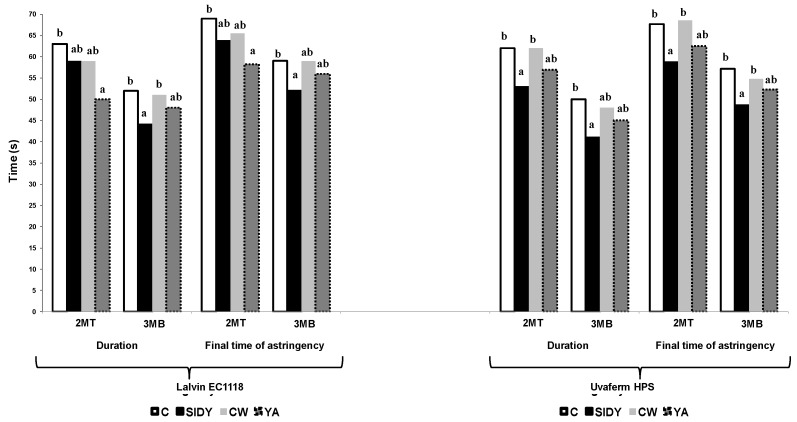
Duration and final time of astringency evaluated in seconds. Columns with different letters indicate statistically significant differences (*p* < 0.05) between the treatments.

**Table 1 molecules-24-01478-t001:** CIELab parameters and low molecular weight phenolic compounds content (mg/L ± SD) of the different red wines studied.

	Lalvin EC1118^®^				Uvaferm HPS^®^			
**2MT**	**CONT**	**SIDY**	**CW**	**YA**	**CONT**	**SIDY**	**CW**	**YA**
**CI**	11.3 ± 0.07 b	9.52 ± 1.07 ab	9.15 ± 1.11 a	10.6 ± 0.06 ab	5.84 ± 0.01 a	5.70 ± 0.02 a	9.41 ± 0.90 c	7.49 ± 0.41 b
**L***	49.3 ± 0.23	55.3 ± 4.29	56.8 ± 4.48	54.3 ± 4.21	71.0 ± 0.29 c	71.7 ± 0.17c	56.0 ± 2.71 a	63.3 ± 0.81 b
**a***	44.0 ± 0.55	38.9 ± 3.60	37.7 ± 2.13	44.1 ± 2.00	31.6 ± 0.55 a	31.3 ± 0.27 a	47.3 ± 2.15 c	40.1 ± 0.49 b
**b***	9.18 ± 0.24	10.0 ± 0.62	10.5 ± 1.68	8.32 ± 0.38	13.3 ± 2.15 b	11.2 ± 0.23 ab	8.4 ± a0.63	8.2 ± 1.54 a
**HBA**	52.2 ± 0.42	51.4 ± 0.22	53.1 ± 0.54	54.5 ± 2.78	44.8 ± 0.07a	45.5 ± 1.08 ab	45.9 ± 0.82 ab	47.2 ± 0.86 b
**HCA**	14.8 ± 0.10	14.6 ± 0.20	14.9 ± 0.07	15.2 ± 0.72	14.1 ± 0.29	14.0 ± 1.21	14.3 ± 0.44	14.9 ± 0.60
**HCATE**	1.89 ± 0.04	1.84 ± 0.07	1.93 ± 0.01	2.09 ± 0.19	4.42 ± 0.06	4.48 ± 0.02	4.42 ± 0.22	4.64 ± 0.12
**TFL**	43.2 ± 0.20 a	44.3 ± 0.59 ab	47.4 ± 0.04 b	45.1 ± 2.51 ab	40.8 ± 1.11 b	42.3 ± 1.02 b	37.5 ± 0.57 a	41.9 ± 1.22 b
**TPRO**	32.9 ± 1.35	33.3 ± 0.61	35.1 ± 0.35	33.3 ± 2.79	29.7 ± 1.69 a	32.2 ± 1.87 ab	32.1 ± 0.50 ab	34.8 ± 0.33 b
**TFLAV**	38.5 ± 1.34	36.4 ± 0.84	38.4 ± 0.14	38.2 ± 2.89	29.7 ± 0.38 a	30.6 ± 1.46 a	33.2 ± 1.72 b	32.8 ± 0.71 b
**TSTILB**	5.13 ± 0.09	4.86 ± 0.16	5.50 ± 0.15	5.38 ± 0.33	5.07 ± 0.05	5.09 ± 0.08	5.73 ± 0.30	5.54 ± 0.20
**TALC**	16.6 ± 0.02	16.4 ± 0.13	17.1 ± 0.15	17.7 ± 0.88	22.2 ± 1.24	21.9 ± 0.76	22.5 ± 1.11	23.2 ± 0.01
**3MB**	**CONT**	**SIDY**	**CW**	**YA**	**CONT**	**SIDY**	**CW**	**YA**
**CI**	11.1 ± 0.34 ab	9.90 ± 0.41 a	9.83 ± 1.08 a	13.1 ± 0.95 b	6.45 ± 0.34 a	6.06 ± 0.02 a	11.2 ± 0.51 c	8.73 ± 0.66 b
**L**	51.4 ± 0.73 ab	54.7 ± 1.47 b	55.6 ± 4.30 b	46.0 ± 2.58 a	69.1 ± 1.02 c	71.0 ± 0.24 c	51.7 ± 1.22 a	60.4 ± 2.84 b
**a**	43.1 ± 0.84 ab	40.3 ± 1.06 ab	39.2 ± 4.19 a	45.7 ± 1.31 b	34.4 ± 1.48 a	32.2 ± 0.35 a	46.4 ± 0.56 c	41.8 ± 0.20 b
**b**	13.7 ± 0.31 b	12.6 ± 0.04 a	13.0 ± 0.18 ab	13.2 ± 0.57 ab	11.8 ± 0.22	13.0 ± 1.51	14.3 ± 0.54	13.8 ± 3.21
**HBA**	46.1 ± 1.56 b	45.7 ± 0.57 b	45.7 ± 0.02b	41.2 ± 2.45 a	40.6 ± 0.83	43.0 ± 2.87	39.3 ± 2.40	42.8 ± 0.04
**HCA**	14.0 ± 0.74 b	13.7 ± 0.05 b	13.2 ± 0.62 ab	11.9 ± 0.41 a	14.1 ± 0.47 a	15.7 ± 0.27 b	14.4 ± 0.26 a	15.2 ± 0.31 ab
**HCATE**	2.49 ± 0.07	2.29 ± 0.13	2.25 ± 0.01	2.16 ± 0.08	3.95 ± 0.20 ab	4.46 ± 0.13 b	2.83 ± 0.02 a	4.48 ± 0.16 b
**TFL**	40.1 ± 2.10 b	40.8 ± 2.00 b	40.2 ± 1.83 b	30.5 ± 1.73 a	30.8 ± 1.81 ab	39.4 ± 2.03 c	28.0 ± 0.96 a	35.6 ± 2.08 b
**TPRO**	31.2 ± 1.85 b	31.2 ± 1.58 b	28.2 ± 1.05 ab	26.0 ± 0.69 a	26.4 ± 1.62 b	29.6 ± 0.35 c	20.1 ± 0.12 a	26.8 ± 1.23 b
**TFLAV**	34.7 ± 1.07 d	32.7 ± 0.57 c	30.9 ± 0.21 b	26.5 ± 0.20 a	23.5 ± 0.22	26.6 ± 1.23	23.6 ± 1.74	25.4 ± 0.48
**TSTILB**	3.64 ± 0.14	3.66 ± 0.01	3.56 ± 0.22	3.21 ± 0.07	5.21 ± 0.08	5.57 ± 0.29	5.05 ± 0.34	5.50 ± 0.09
**TALC**	16.4 ± 0.64 b	16.6 ± 0.20 b	16.8 ± 0.19 b	14.5 ± 0.01 b	19.7 ± 0.10 a	21.5 ± 1.55 ab	20.3 ± 1.03 a	22.6 ± 0.37 b

Values in the same row with different letters indicate statistically significant differences (*p* < 0.05). 2MT (two months of treatment). 3MB (three months of bottle storage). HBA: hydroxybenzoic acids; HCA: hydroxycinnamic acids; HCATE: hydroxycinnamic acid tartaric esters; TFL: total flavanol monomers; TPRO: total proanthocyanidins; TFLAV: total flavonols; TSTILB: total stilbenes; TALC: total alcohols.

**Table 2 molecules-24-01478-t002:** Volatile compound composition in red wines fermented with Lalvin EC1118^®^ (average ± standard deviation) expressed in µg/L.

	LRI	ID	AMLF	2MT				3MB			
			C	C	SIDY1	CW	YA	C	SIDY1	CW	YA
**Ethyl esters**											
Ethyl butanoate	1076	A	178 ± 13	144 ± 9	171 ± 10	159 ± 22	156 ± 7	138 ± 19a	158 ± 90a	204 ± 2b	201 ± 2b
Ethyl hexanoate	1246	A	291 ± 43	214 ± 4	254 ± 27	230 ± 64	199 ± 3	198 ± 34a	225 ± 32a	359b ± 3	375 ± 24b
Ethyl heptanoate	1334	B	10.9 ± 0.90	10.5 ± 1.20	12.7 ± 0.80	10.3 ± 1.20	11.3 ± 0.20	7.87 ± 0.62a	8.92a ± 0.69	10.7b ± 0.30	11.3b ± 0.60
Ethyl lactate	1413	A	19.9 ± 1.60	29.9 ± 8.40	38.2 ± 7.30	42.1 ± 1.20	38.9 ± 4.2	19.2 ± 1.70a	20.8 ± 3.50ab	31.0 ± 6.30b	28.4 ± 1.50b
Ethyl octanoate	1460	A	888 ± 83	739 ± 12	782 ± 59	728 ± 95	794 ± 69	629 ± 111a	720 ± 7a	998 ± 15b	1096 ± 152b
Ethyl nonanoate	1558	A	45.6 ± 2.7	27.0 ± 3.50	36.0 ± 0.10	32.5 ± 12.70	28.1 ± 6.50	17.9 ± 0.60a	21.7 ± 0.30ab	22.9 ± 1.10b	24.1 ± 3.60b
Ethyl succinate	1701	A	92 ± 5	108 ± 2	111 ± 9	113 ± 23	114 ± 11	134 ± 9a	142 ± 2a	176 ± 10b	195 ± 7b
Ethyl decanoate	1715	A	127 ± 7	51.0 ± 3.80a	52.6 ± 5.30ab	47.7 ± 2.0a	60.7 ± 5.60b	34.5 ± 7.80a	47.1 ± 0.30ab	61.8 ± 4.10b	70.9 ± 21b
Ethyl isovalerate	1806	A	4.09 ± 0.08	2.64 ± 1.36	2.82 ± 1.90	5.55 ± 2.19	6.27 ± 2.94	1.50 ± 0.210	3.36 ± 1.02	2.09 ± 0.34	3.18 ± 3.02
Ethyl undecanoate	1824	A	1.20 ± 0.10	0.876 ± 0.06	1.07 ± 0.09	1.07 ± 0.27	1.07 ± 0.08	0.680 ± 0.014a	0.745 ± 0.05a	0.810 ± 0.03a	1.01 ± 0.07b
Ethyl dodecanoate	1869	B	116 ± 8	31.5 ± 1.0	38.3 ± 3.30	32.3 ± 2.50	39.5 ± 12.40	14.5 ± 2.10ab	10.5 ± 2.10a	20.2 ± 0.20b	20.4 ± 5.60b
Ethyl tetradecanoate	2068	B	16.0 ± 2.10	9.90 ± 0.53	11.6 ± 1.10	11.1 ± 0.20	11.1 ± 1.30	8.26 ± 0.01a	7.02 ± 1.62a	12.1 ± 0.60b	9.37 ± 0.87a
Ethyl hexadecanoate	< 2100	B	21.6 ± 1.80	14.5 ± 0.80	15.8 ± 0.10	17.2 ± 2.0	14.1 ± 1.10	11.5 ± 0.30a	11.1 ± 2.0a	14.7 ± 3.80b	16.6 ± 0.70b
**Methyl esters**											
Methyl hexanoate	1183	A	1.63 ± 0.23	0.99 ± 0.28	1.45 ± 0.25	1.45 ± 0.16	1.36 ± 0.02	0.81a ± 0.36	1.27ab ± 0.07	2.00b ± 0.25	2.09b ± 0.50
Methyl octanoate	1420	A	6.80 ± 0.66	5.10 ± 0.12	5.65 ± 0.32	5.26 ± 0.86	6.40 ± 0.59	4.69 ± 0.50a	4.55 ± 0.69a	6.52 ± 0.06ab	7.82 ± 1.52b
Methyl decanoate	1632	A	2.06 ± 0.25	nd	nd	nd	nd	nd	nd	nd	nd
**Acetate esters**											
Isoamyl acetate	1163	A	2117 ± 253	584 ± 30a	1472 ± 306b	1572 ± 110b	884 ± 170a	706 ± 291a	939 ± 61a	1872 ± 90b	1084 ± 163a
Hexyl acetate	1306	A	2.39 ± 0.07	1.73 ± 0.10	1.92 ± 0.02	1.78 ± 0.00	1.92 ± 0.16	2.01 ± 0.40	1.68 ± 0.26	2.30 ± 0.05	2.82 ± 0.76
2-phenylethyl acetate	1851	A	139 ± 34	64.9 ± 2.8	68.5 ± 3.20	61.4 ± 2.70	73.5 ± 10.80	121 ± 42ab	80.6 ± 14.40a	91.4 ± 16.70a	174 ± 15b
**Isoamyl esters**											
Isopentyl hexanoate	1478	A	0.740 ± 0.01	0.581 ± 0.03	0.605 ± 0.09	0.471 ± 0.195	0.597 ± 0.05	0.423 ± 0.185a	0.526 ± 0.09a	0.843 ± 0.045b	0.898 ± 0.06b
Isoamyl octanoate	1748	A	1172 ± 110	1152 ± 1	1252 ± 171	1055 ± 168	955 ± 64	662a ± 147	958b ± 17	1238c ± 45	1205c ± 97
Isoamyl decanoate	1909	A	138 ± 10	44.5 ± 2.0	48.2 ± 3.80	40.0 ± 2.70	51.8 ± 3.30	16.4 ± 5.10	12.7 ± 0.80	20.9 ± 2.0	28.2 ± 8.2
**Alcohols**											
Isobutanol	1108	A	70577 ± 4434	60577 ± 1383	65910 ± 4970	63910 ± 11482	62910 ± 2805	64243 ± 6881a	68910 ± 1391a	89243 ± 6641b	90243 ± 2579b
3-Methyl-1-butanol	1197	A	107593 ± 6486	98038 ± 396	106260 ± 5796	107593 ± 3115	99704 ± 2800	97816 ± 7230a	104593 ± 4162a	134927 ± 3090b	133816 ± 3935b
Hexanol	1391	A	2377 ± 177	2227 ± 10	2477 ± 193	2377 ± 134	2327 ± 24	2402 ± 154a	2577 ± 170a	3452 ± 247b	3502 ± 76b
Heptanol	1478	B	29.4 ± 1.30	30.4 ± 1.0	32.9 ± 1.90	31.9 ± 2.60	31.4 ± 1.50	31.9 ± 0.40a	34.4 ± 2.0a	43.4 ± 3.10b	43.9 ± 1.90b
Octanol	1578	A	7.24 ± 0.43	8.61 ± 0.01	9.72 ± 1.20	9.26 ± 1.30	9.17 ± 0.45	9.44 ± 0.15a	10.3 ± 0.40a	14.3 ± 0.60b	14.5 ± 0.10b
Decanol	1773	A	3.09 ± 0.29	3.32 ± 0.02	3.56 ± 0.08	3.50 ± 0.29	3.44 ± 0.06	3.50 ± 0.00	3.74 ± 0.13	4.68 ± 0.02	4.74 ± 0.13
Benzyl alcohol	1978	B	360 ± 18	618 ± 16	675 ± 79	650 ± 164	643 ± 82	658 ± 114	705 ± 171	910 ± 334	925 ± 102
2-Phenylethanol	2020	A	15689 ± 731	14329 ± 173a	15829 ± 2002b	15209 ± 3789ab	14989 ± 1973a	15349 ± 2347a	16609 ± 176a	22109 ± 421b	22509 ± 2191b
**Volatile fatty acids**											
Hexanoic acid	1894	A	1170 ± 20	1200 ± 15	1268 ± 135.60	1385 ± 250	1288 ± 118	1168 ± 125a	1290 ± 28ab	1568 ± 88bc	1660 ± 151c
Octanoic acid	< 2100	A	209 ± 19	113 ± 45	203 ± 98.90	304 ± 125	284 ± 80	124 ± 109a	243 ± 37ab	364 ± 22b	534 ± 26c
Decanoic acid	< 2100	A	102 ± 1	42.0 ± 0.5	42.8 ± 3.0	46.8 ± 4.80	66.8 ± 15.50	41.6 ± 3.90a	45.6 ± 3.60ab	52.4 ± 6.90ab	75.6 ± 22.4b
Dodecanoic acid	< 2100	B	14.04 ± 0.84	14.84 ± 1.46	16.1 ± 0.10	16.8 ± 1.0	14.84 ± 0.48	15.24 ± 0.12	14.84 ± 0.62	16.4 ± 1.40	16.04 ± 0.31
**Terpenes**											
Citronellol	1785	A	3.44 ± 0.09	3.60 ± 0.01	3.76 ± 0.50	3.76 ± 0.44	3.60 ± 0.10	3.76 ± 0.12a	3.92 ± 0.10a	4.87 ± 0.00b	5.03 ± 0.13b
Nerol	1887	A	16.6 ± 1.0	24.3 ± 1.10	25.7 ± 1.40	25.2 ± 3.10	24.8 ± 2.50	25.2 ± 0.70a	26.6 ± 0.70a	32.5 ± 2.40b	33.0 ± 0.30b
trans-nerolidol	2056	A	2.61 ± 0.09	3.66 ± 0.18	3.95 ± 0.14	3.86 ± 0.16	3.86 ± 0.08	3.86 ± 0.07a	4.05 ± 0.08a	4.91 ± 0.06b	5.01 ± 0.15b

Values with different letter in the same row, and within of each point of analysis, indicate statistically significant differences (*p* < 0.05). Values without letter indicate no statistically significant differences. AAF: After alcoholic fermentation, 2MT: two months of treatment, 3MB: three months of bottle storage. LRI: Linear Retention Index. ID: reliability of identification: A, mass spectrum and LRI agreed with standards; B, mass spectrum agreed with mass spectral data base and/or LRI agreed with the literature data.

**Table 3 molecules-24-01478-t003:** Volatile compounds composition in the red wines fermented with Uvaferm HPS^®^ (average ± standard deviation) expressed in µg/L.

	LRI	ID	AMLF	2MT				3BS			
			C	C	SIDY1	CW	YA	C	SIDY1	CW	YA
**Ethyl esters**											
Ethyl butanoate	1076	A	234 ± 10	203 ± 8a	214 ± 7ab	203 ± 12a	232 ± 2b	192 ± 16	213 ± 1	207 ± 5	199 ± 1
Ethyl hexanoate	1246	A	396 ± 29	310 ± 32ab	325 ± 7ab	292 ± 49a	392 ± 17b	304 ± 28a	346 ± 13b	327 ± 19ab	303 ± 13a
Ethyl heptanoate	1334	B	9.57 ± 0.21	9.24 ± 1.89	8.33 ± 1.08	9.37 ± 1.78	9.96 ± 0.42	7.41 ± 0.45	7.80 ± 0.43	8.26 ± 0.32	7.61 ± 0.04
Ethyl lactate	1413	A	3.14 ± 1.17	2.86 ± 0.21	2.82 ± 1.04	1.75 ± 0.73	3.12 ± 0.55	10.5 ± 2.8c	6.64 ± 2.73ab	14.5 ± 1.7c	3.40 ± 0.04a
Ethyl octanoate	1460	A	1143 ± 47	928 ± 117a	1049 ± 17ab	826 ± 107a	1229 ± 120b	786 ± 61a	1069 ± 96b	853 ± 70ab	924 ± 79ab
Ethyl nonanoate	1558	A	56.7 ± 0.9	31.7 ± 5.3	26.6 ± 4.5	23.8 ± 0.7	28.8 ± 9.8	14.3 ± 2.0	15.4 ± 2.0	17.7 ± 1.7	17.9 ± 1.0
Ethyl succinate	1701	A	333 ± 6	172 ± 5a	345 ± 1b	132 ± 11a	297 ± 2b	159 ± 7a	278 ± 7b	170 ± 6a	267 ± 4b
Ethyl decanoate	1715	A	22.1 ± 2.4	36.4 ± 2.2b	29.7 ± 3.8a	36.1 ± 2.4b	38.1 ± 29.3b	54.8 ± 3.8ab	54.0 ± 14.2ab	59.7 ± 0.6b	48.9 ± 7.7a
Ethyl undecanoate	1824	A	0.680 ± 0.030	0.549 ± 0.016b	0.614 ± 0.042c	0.418 ± 0.021a	0.484 ± 0.133a	2.18 ± 0.04	3.16 ± 0.05	2.44 ± 0.01	2.05 ± 0.07
Ethyl dodecanoate	1869	B	14.3 ± 1.0	6.95 ± 0.17a	20.5 ± 0.4c	5.25 ± 0.22a	12.9 ± 2.6b	22.6 ± 1.4a	34.1 ± 1.5b	23.9 ± 0.9a	24.9 ± 1.8a
Ethyl tetradecanoate	2068	B	16.0 ± 1.1	7.54 ± 0.01a	9.83 ± 0.27bc	9.05 ± 0.52ab	11.1 ± 1.1c	16.7 ± 2.5b	14.8 ± 1.5b	18.3 ± 1.8b	7.35 ± 0.52a
Ethyl hexadecanoate	< 2100	B	28.3 ± 4.2	14.6 ± 0.8a	21.3 ± 0.4c	18.1 ± 1.2b	23.4 ± 0.2c	16.4 ± 1.3a	29.7 ± 1.8b	16.7 ± 0.2a	13.4 ± 1.0a
**Methyl esters**											
Methyl hexanoate	1183	A	1.63 ± 0.27	1.17 ± 0.11a	1.27 ± 0.02a	1.08 ± 0.21a	1.72b ± 0.00	1.08 ± 0.20	1.17 ± 0.47	1.08 ± 0.32	0.807 ± 0.167
Methyl octanoate	1420	A	6.64 ± 0.29	4.33 ± 0.86a	4.98 ± 0.37ab	4.09 ± 0.83a	6.54 ± 1.35b	3.64 ± 0.51a	5.42 ± 0.41b	4.02 ± 0.41a	4.31 ± 0.55ab
Methyl decanoate	1632	A	2.72 ± 0.03	nd	1.46 ± 0.17	nd	1.26 ± 1.07	nd	1.39 ± 0.44	nd	nd
**Acetate esters**											
Isoamyl acetate	1163	A	5928 ± 400	4350 ± 560a	5328 ± 123b	4261 ± 121a	5717 ± 149b	3572 ± 111a	4695 ± 412b	3350 ± 292a	3750 ± 189a
Hexyl acetate	1306	A	8.22 ± 0.43	4.38 ± 1.83a	21.6 ± 0.4b	4.86 ± 0.51a	14.6 ± 8.1ab	3.63 ± 0.01a	17.3 ± 2.0b	3.01 ± 0.05a	8.13 ± 1.02ab
2-Phenylethyl acetate	1851	A	1069 ± 37	285 ± 21a	869 ± 47c	246 ± 11a	604 ± 56b	188 ± 48a	309 ± 8b	138 ± 15a	201 ± 36a
**Isoamyl esters**											
Isoamyl hexanoate	1478	A	1.53 ± 0.07	1.31 ± 0.25	1.07 ± 0.07	1.14 ± 0.22	1.51 ± 0.19	0.922 ± 0.104	1.19 ± 0.10	1.10 ± 0.16	0.977 ± 0.079
Isoamyl octanoate	1748	A	1898 ± 79	2008 ± 21b	2422 ± 29c	1362 ± 12a	2155 ± 155b	1338 ± 37a	1802 ± 44b	1385 ± 87a	1685 ± 163b
Isoamyl decanoate	1909	A	533 ± 65	595 ± 24a	571 ± 6a	673 ± 77ab	743 ± 19b	44.5 ± 5.3a	96.4 ± 1.0c	38.2 ± 4.5a	74.5 ± 5.7b
**Alcohols**											
Isobutanol	1108	A	64910 ± 4893	61577 ± 1415	60910 ± 6598	54243 ± 2798	62910 ± 94	59243 ± 4028	58910 ± 4529	63910 ± 3131	58910 ± 3525
3-Methyl-1-butanol	1197	A	136038 ± 6490	128260 ± 953a	129371 ± 5136a	123816 ± 6513a	142704 ± 3344b	134927 ± 1213	140482 ± 9572	134927 ± 3572	128260 ± 1694
Hexanol	1391	A	2402 ± 141	2252 ± 4ab	2252 ± 131ab	2152 ± 136a	2502 ± 102b	2402 ± 46	2477 ± 218	2377 ± 91	2252 ± 54
Heptanol	1478	B	20.9 ± 0.5	19.9 ± 0.3	18.9 ± 1.1	19.9 ± 0.9	21.4 ± 0.8	21.4 ± 0.1	20.9 ± 1.7	21.9 ± 1.0	19.9 ± 0.1
Octanol	1578	A	9.17 ± 0.85	8.06 ± 0.33a	8.71 ± 0.65a	8.61 ± 0.45a	10.2 ± 0.3b	9.17 ± 0.17	9.62 ± 0.56	8.98 ± 0.68	8.89 ± 0.03
Decanol	1773	A	3.38 ± 0.22	2.79 ± 0.06	2.97 ± 0.11	2.91 ± 0.21	3.21 ± 0.22	2.97 ± 0.05	3.03 ± 0.14	2.74 ± 0.03	2.74 ± 0.03
Benzyl alcohol	1978	B	265 ± 47	220 ± 4	220 ± 18	215 ± 21	240 ± 19	260 ± 17	268 ± 24	283 ± 40	213 ± 2
2-Phenylethanol	2020	A	24709 ± 4375	20509 ± 30	21109 ± 1625	20909 ± 2112	23709 ± 1537	21309 ± 452ab	26109 ± 2628b	23309 ± 2543ab	20509 ± 38a
**Volatile fatty acids**											
Hexanoic acid	1894	A	1885 ± 262	1695 ± 16	1785 ± 132	1707 ± 179	1932 ± 120	1810 ± 7ab	2070 ± 128b	1852 ± 158ab	1760 ± 23a
Octanoic acid	< 2100	A	1074 ± 246	584 ± 46a	1054 ± 185b	634 ± 116a	1124 ± 15b	754 ± 58a	1354 ± 96b	724 ± 87a	864 ± 49a
Decanoic acid	< 2100	A	166 ± 12	56.8 ± 2.0a	138 ± 5b	60.0 ± 5.8a	135 ± 10b	69.2 ± 0.6a	150 ± 11c	74.0 ± 3.7a	112 ± 9b
Dodecanoic acid	< 2100	B	18.1 ± 0.4	16.8 ± 0.3	16.4 ± 0.3	15.6 ± 0.8	16.4 ± 0.3	14.8^b^ ± 0.2	18.1^d^ ± 0.7	16.4^c^ ± 0.4	12.4^a^ ± 0.4
**Terpenes**											
Citronellol	1785	A	4.56 ± 0.24	4.24 ± 0.05	4.40 ± 0.46	4.40 ± 0.32	4.87 ± 0.22	4.24 ± 0.24	4.56 ± 0.34	4.24 ± 0.12	4.08 ± 0.29
Nerol	1887	A	13.9 ± 1.5	12.5 ± 0.3	11.6 ± 0.5	13.4 ± 1.4	13.9 ± 0.5	12.0 ± 0.7	12.0 ± 0.6	11.6 ± 0.1	11.6 ± 0.2
*trans*-Nerolidol	2056	A	2.51 ± 0.08	2.80 ± 0.24ab	2.32 ± 0.08a	3.18 ± 0.03b	2.22 ± 0.33a	3.09 ± 0.49a	2.61 ± 0.23a	4.63 ± 0.01b	2.41 ± 0.18a

Values with different letter in the same row, and within of each point of analysis, indicate statistically significant differences (*p* < 0.05). Values without letter indicate no statistically significant differences. AMLF: After malolactic fermentation, 2MT: two months of treatment, 3MB: three months of bottle storage. LRI: Linear Retention Index. ID: reliability of identification: A, mass spectrum and LRI agreed with standards; B, mass spectrum agreed with mass spectral data base and/or LRI agreed with the literature data.

**Table 4 molecules-24-01478-t004:** Classic oenological parameters of red wines after malolactic fermentation.

	Lalvin EC1118^®^	Uvaferm HPS^®^
Alcoholic degree (vol %)	13.8	13.7
pH	3.91	3.82
AT (g/L)	3.2	3.4
AV (g/L)	0.41	0.38

TA: Total acidity expressed in g/L of sulphuric acid. VA: Volatile acidity, expressed in g/L of acetic acid.

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
