# Peer review of "Evaluation of Yeast Derivative Products Developed as an Alternative to Lees: The Effect on the Polysaccharide, Phenolic and Volatile Content, and Colour and Astringency of Red Wines"

_molecules, 2019, doi:10.3390/molecules24081478_

Round 1

Reviewer 1 Report

Dear Authors,

I recommend the publication of the manuscript with minor changes considering the followings:

English language proofing is needed

there are some fragments which have to be translated in English (lines 305-306)

some cited references are older than 2010 (more than 40% of the cited references); I suggest to partially replace them.

Author Response

Dear reviwer, we have applied the minor changes suggested:

 -         The english grammar mistakes were revised, and all the manuscript was sent to an English editing service checking grammar.

-         Lines 305- 306: The fragments were translated to english.

-         Some cited references were replaced by others more current ones.

Reviewer 2 Report

The manuscript entitled “Evaluation of yeast derivative products developed as an alternative to lees: effect on the polysaccharide, phenolic and volatile content, colour and astringency of red wines” deals with the study of the chemical impact of the use of different preparations of yeast derivatives in the wine composition. In my opinion, it provides interesting results to the field and makes a good and objective discussion of the existing knowledge in relation to the subject under study. However, it is not possible to publish it in its current form, it needs to improve certain aspects, especially those related to English and grammar. I highly recommend a thorough revision of the text by a native English speaker since there are many mistakes regarding verbal tenses, as well as spelling and grammar mistakes, see for example line 17 (correct the grammar and the verbal tense of “in this work was evaluate the application effect”), line 27 (replace “depended of” with “depended on”), line 123 (stables should be in singular as it is an adjective), line 136 (the order of the sentence is wrong), line 141 (“degradation of in the control wines” of what?), line 155 (“this results” is singular and “are” is plural; it does not match), line 196 (“matherial” should be replaced by “material”, the same mistake is repeated later), line 378 (correct the verbal tense: “not produced”), line 391 (correct the verbal tense: “could well correlated”)…

In addition, the following things must be revised before acceptance:

Abstract:

Line 20: Indicate the dosage used of the tested YDs.

Line 27: Show the full meaning of YDs, at least the first time it is mentioned.

Introduction:

Line 34: How can YDs improve the technological processes? Include it to better explain its actual usefulness.

Line 37: Specify here which are the main disadvantages.

Line 90: Where are the results of the 2016 harvest?

Results and discussion:

Line 97: Replace “harvests” with “harvest”

Figure 1: Include the legend for the different polysaccharides fractions, F1, F2 and F3. Show the SD values in the bars. Indicate the statistical meaning in the figure caption. And the values that are represented in the figure, for example: means ± SD.

Subsection 2.2: Rename it as “Effect of YDs application on…”

Line 121: Reference to Table 2 is mistaken. Please, correct, should be Table 1.

Line 123: “Stables” should be in singular form as it is an adjective

Figure 2: Show the SD in the bars

Line 136: Rewrite the full sentence

Line 137: Replace “depending of” with “depending on”

Line 138: Correct the sentence, since the subject is in singular and the verb in plural

Line 141: Correct the sentence: “degradation of ¿? in the control wines”.

Line 150: Correct the sentence, the grammar is not ok

Subsection 2.3: Rename it as “Effect of YDs application on…”

Line 155: Replace “This result” with “These results” if plural

Line 163: What is the meaning of “colour stability or colour”??

Table 1: Include “Values in the same row with different…” in the caption. Include the SD values in the table and show the data as mean ± SD.

Tables 2 and 3: Use the same decimal places for all the data set.

Line 189: Which content? Total volatiles?

Line 192: Correct the sentence: “concentration esters of isoamyl and acetate esters”

Line 196: “3BS”?? I suppose it should be “3MB”? as it is described in M&M section… Same in lines 214, 223 and 228.

Figure 3: Include the description of the statistical results in the caption of the figure. In the legend, I would write “YA” instead of “Yeast autolysate” to keep the same nomenclature. Same for figure 4.

Figure 5: Resolution is too low. Try to improve the total DPI (dots per inch) of the figure, a value of 300 DPI should be enough. Include the units in the caption.

Line 273: Correct the sentence: “statistical differences were found” it is repeated.

Materials and methods:

There are several words in Spanish that should be translated (See lines 294 “y”, 298 “de”, 299 “de”, 304 “Tabla”, The caption of table 4 and the row label “grado alcohólico”, 338 “de”).

Line 307: ppm should be replaced by mg/L since they are the units accepted in the international system

Line 308: Why have you selected plastic as the material for the ageing treatments?

Line 318: Please, indicate the filtration conditions.

Line 336: Please, provide the full name of the technique HPSEC-RID, at least the first time mentioned.

Line 337: “Polypehol” should be corrected

Line 344: “development” should be replaced by “developed”

Once these questions are corrected, I suggest its acceptance in molecules.

Author Response

Dear reviewer, here we have detailed the changes suggested:

The english grammar mistakes were revised, and all the manuscript was sent to an English editing service checking grammar. Here we have responsed to all the comments of the reviewer:

Line 17: the verbal tense as corrected

Line 27: It was replaced.

Line 123: Stables was changed by stable.

Line 136: The sentence was rewrote.

Line 41: (“degradation of in the control wines” of what?): phenolic compounds.

Line 155: It was whote in the correct form.

Line 196: Matherial was replace by material.

Line 378: The verbal tense was corrected.

Line 391: The verbal tense was corrected.

Abstract:

Line 20: The dosage used of the tested YDs was indicated.

Line 27: The full meaning of YDs was mentioned in line 16.

Introduction:

Line 34: The technological processes were replaced by technological characteristics and these characteristics were included.

Line 37: The most important disadvantage was mentioned.

Line 90: It was deleted because was specify in the part of winemaking and experimental design (only data of 2015 harvest were used).

Results and discussion:

All the changes suggested by the reviewer were applied.

Materials and methods:

All the changes suggested by the reviewer were applied.

Line 308: Why have you selected plastic as the material for the ageing treatments?: Plastic thanks were selected because were easier to move the wine from the winery to the University pilot plant. And because it was easier for number of thanks for replications.